# High-fidelity musculoskeletal modeling reveals that motor planning variability contributes to the speed-accuracy tradeoff

**Mazen Al Borno[1,2]\*, Saurabh Vyas[1], Krishna V Shenoy[1,3,4,5,6,7], Scott L Delp[1,8]**

[1]Department of Bioengineering, Stanford University, Stanford, United States; [2]Department of Computer Science and Engineering, University of Colorado Denver, Denver, United States; [3]Neurosciences Program, Stanford University, Stanford, United States; [4]Department of Electrical Engineering, Stanford University, Stanford, United States; [5]Wu Tsai Neuroscience Institute, Stanford University, Stanford, United States; [6]Department of Neurobiology, Stanford University, Stanford, United States; [7]Howard Hughes Medical Institute, Stanford University, Stanford, United States; [8]Department of Mechanical Engineering, Stanford University, Stanford, United States

**Abstract** A long-standing challenge in motor neuroscience is to understand the relationship between movement speed and accuracy, known as the speed-accuracy tradeoff. Here, we introduce a biomechanically realistic computational model of three-dimensional upper extremity movements that reproduces well-known features of reaching movements. This model revealed that the speed-accuracy tradeoff, as described by Fitts' law, emerges even without the presence of motor noise, which is commonly believed to underlie the speed-accuracy tradeoff. Next, we analyzed motor cortical neural activity from monkeys reaching to targets of different sizes. We found that the contribution of preparatory neural activity to movement duration (MD) variability is greater for smaller targets than larger targets, and that movements to smaller targets exhibit less variability in population-level preparatory activity, but greater MD variability. These results propose a new theory underlying the speed-accuracy tradeoff: Fitts' law emerges from greater task demands constraining the optimization landscape in a fashion that reduces the number of 'good' control solutions (i.e., faster reaches). Thus, contrary to current beliefs, the speed-accuracy tradeoff could be a consequence of motor planning variability and not exclusively signal-dependent noise.

**\*For correspondence:**
mazen.alborno@ucdenver.edu

**Competing interests:** The authors declare that no competing interests exist.

## Introduction

Elite tennis players are particularly skilled at striking a balance between the speed and the accuracy of their serves. If they only focus on placing the ball in a strategic location, they may inadvertently advantage their opponent by not paying close enough attention to the speed of their serve. On the other hand, serving the ball with as much speed as possible is not desirable as the ball may fail to clear the net or land in a location that is advantageous to their opponent. This relationship, commonly known as the 'speed-accuracy tradeoff,' is observed not only for purely motor, but also perceptual and cognitive tasks (e.g., *Heitz, 2014*; *Wickelgren, 1977*). In laboratory settings, the speed-accuracy tradeoff, quantified by Fitts' law (*Fitts, 1954*), is an empirical finding that movements that require greater accuracy tend to be slower than those that do not. Fitts' law relates the width of the target (*W*), the movement distance (*A*), and the movement duration (MD):

$$\mathrm{MD} = a + b\log_2(2A/W),$$

where $a$ and $b$ are subject-specific scalar parameters. The $\log_2(2A/W)$ term is known as the 'index of difficulty'.

A widely held theory, initially proposed by Harris and Wolpert, posits that signal-dependent noise (i.e., 'motor noise' present at the neuromuscular junction) that scales with the size of the control signal is a sufficient theory to explain Fitts' law (*Harris and Wolpert, 1998*; *Lunardini et al., 2015*; *McCrea and Eng, 2005*; *Peternel et al., 2017*). The theory proposes that if motor noise increases with the size of the control signal, then moving rapidly, which requires large control signals, increases the variability in the final movement position. Conversely, reaching to a smaller target, with low variability in the final position, requires smaller control signals that consequently produce slower movements. Thus, this theory provides an explanation for the speed-accuracy tradeoff. In the present work, we show that the speed-accuracy tradeoff, as described by Fitts' law, emerges even without the presence of motor noise.

In support of this theory, a large body of prior work has employed simple one-dimensional linear systems or torque-driven two-joint models of the arm that are restricted to moving in a plane to study the speed-accuracy tradeoff (e.g., *Flash and Hogan, 1985*; *Fagg et al., 2002*; *Harris and Wolpert, 1998*; *Alexander, 1997*; *DeWolf et al., 2016*; *Sketch et al., 2017*). While it certainly is not the case that the mechanical properties of the plant being controlled explain phenomena such as Fitts' law, the patterns of neural activity required to drive goal-directed movements are necessarily constrained by the dynamics of the plant. When musculoskeletal dynamics are modeled with sufficient fidelity, and these musculoskeletal dynamics are controlled in an optimal fashion, motor behavior that is consistent with Fitts' law emerges and thus provides a parsimonious theory of the speed-accuracy tradeoff.

Thus, a primary innovation in this work is the development of a computational model of upper extremity movements with a more realistic model of the musculoskeletal system than previously described (three-dimensional, 47 muscles and five degrees-of-freedom [*Saul et al., 2015*]). The musculoskeletal model proposed by Saul and colleagues was previously used to track experimental kinematics, but not synthesize movements de novo. Our computational model synthesizes three-dimensional point-to-point reaching movements that reproduce features reported in motor control studies and in our experimental data. We focus on point-to-point movements because normal voluntary movements can be described as holding still after moving from one pose to another (*Shadmehr, 2017*). The movements are synthesized by minimizing a cost function based on end-point accuracy and squared muscle activation terms. We use the squared muscle activation term as an approximation to metabolic power consumption (*Millard et al., 2013*). We show that the asymmetry in the velocity profile with movement speed can be explained in terms of optimal control theory, as opposed to it either representing some unexplained feature, or resulting from constraints in the neural architecture (*Bullock and Grossberg, 1988*), or a learned strategy when approaching a target (*Beggs and Howarth, 1972*), or time-varying cost constraints (*Nagasaki, 1989*), or signal-dependent noise with a minimum variance model (*Tanaka et al., 2004*).

Guided by this computational model, we propose a new theory based on motor planning variability: Fitts' law emerges as a consequence of the difficulty associated with finding good control solutions for challenging tasks. Concretely, our theory results from the observation that during learning of a new task, the search space for finding the control solutions is very high-dimensional and contains many local minima. In our model, we use a stochastic optimizer, which attempts to find appropriate motor plans (i.e., reach trajectories) in this search space. As a consequence of the high dimensionality of the search space and the presence of local minima, the optimizer is more likely to find locally optimal solutions (or solutions that are near locally optimal solutions) in lieu of globally optimal solutions. Our theory suggests that for more challenging tasks (e.g., reaching to smaller targets), the trajectory optimizer tends to find less effective solutions (i.e., slower reaches) than other undiscovered but possible solutions. For less challenging tasks (e.g., reaching to larger targets), the optimizer tends to find more effective solutions (i.e., faster reaches). With more challenging tasks, previously effective solutions are discarded as they do not produce trajectories that reach the target; the smaller set of effective solutions makes it more likely for the optimizer to converge to a less effective solution.

The crux of our theory is that some features of human movement are attributable to planning variability rather than execution noise (e.g., Figure 4 and *Churchland et al., 2006a* for related evidence). We analyze motor cortical neural activity from monkeys reaching to targets of different sizes. We hypothesized that if motor planning variability has a contribution to the speed-accuracy tradeoff, then we should observe a statistically significant difference in motor planning variability when reaching to large and small targets; this is not expected if the speed-accuracy tradeoff is only the result of execution noise. Furthermore, we hypothesized that the contribution of planning variability to MD variability should increase for smaller targets (i.e., as sensitivity to the motor plan increases for more challenging tasks). We test both hypotheses in this work and show that the neural analyses are consistent with our theory.

A realistic musculoskeletal model allows us to study human movement variability with greater resolution. Our theory highlights the role of motor planning rather than execution noise to explain features of human movement. That is not to say that signal-dependent noise does not exist or could not play a part in Fitts' law. Rather, given the results here, we must re-examine the assumption that signal-dependent noise is the only factor that gives rise to Fitts' law. Reaching to a smaller target may exhibit smaller velocities to minimize the effects of signal-dependent noise. Alternatively, slower reach trajectories for smaller targets might be 'easier' to find from an optimal control theory perspective. We believe our new theory provides a more complete account of the empirical observations, both behaviorally and neurally, surrounding the speed-accuracy tradeoff, without having to rely on assumptions surrounding improvements in the signal-to-noise ratio.

## Results

### Model validation

We first evaluate whether the computational model reproduces important kinematic features reported in motor control studies. In *Figure 1A and B*, we compare the hand positions in a center-out fast reaching task from our model and the data reported in *Beer et al., 2000*. The simulations are synthesized with trajectory optimization (see the section Trajectory optimization), specifying only the final target position and the MD, which we set to 350 ms. Note that our simulations reproduce the observation that arm reach trajectories (in humans and non-human primates) are gently curved, particularly near the targets. In *Figure 1C and D*, we compare the velocity profile of our simulation with the data presented in *Soechting, 1984*. The maximum unnormalized speed in *Figure 1D* is 1.10 m/s, which falls in the range of maximal speeds reported in *Soechting, 1984* (i.e., from 0.65 m/s to 1.35 m/s). We also compare the effect of the target size on the synthesized and experimental data in *Figure 2A and B*. Soechting reported that a smaller target causes the movement to have an earlier peak velocity and a velocity that decays more rapidly. We reproduce both of these features in our simulations. However, the experimental speed profiles of the large and small targets are more nearly parallel than the simulated ones. The areas of the small and large targets are 2.55 cm$^2$ and 3.54 cm$^2$. The velocity profiles of our computational model are determined from an average of 15 runs.

Several studies have shown that the velocity profile is symmetric for intermediate speeds and becomes asymmetric with changes in speed (e.g., *Nagasaki, 1989*; *Ostry et al., 1987*). The ratio of the time to the peak velocity to the entire movement time tends to be smaller with slower movements. To study this, we optimize for 20 fast (0.18 s MD), medium (0.22 s MD), and slow (0.26 s MD) reaches to the same target. In *Figure 2C*, we compare the normalized velocity profiles (with normalized time) between the best solutions (i.e., the trial with the lowest cost in *Equation 1* in Materials and methods) for the movements at different speeds. Our model reproduces the speed-dependent asymmetry in the velocity profile. The relative peak velocities in *Figure 2C* are 0.66, 0.51, and 0.4 for the fast, medium, and slow movements, respectively. It has also been reported that the velocity profiles of slower movements tend to exhibit multiple local minima (e.g., *Soechting, 1984*; *van der Wel et al., 2009*); we replicate these findings in our simulations (see *Figure 2—figure supplement 1* for the velocity profile asymmetry for MDs of 0.15 s, 0.25 s, and 0.45 s). We perform the same experiment with a torque-driven model for MDs of 0.2 s, 0.25 s, and 0.3 s. In *Figure 2—figure supplement 2*, we show that this simplified model does not reproduce the reported asymmetries in the velocity profiles.

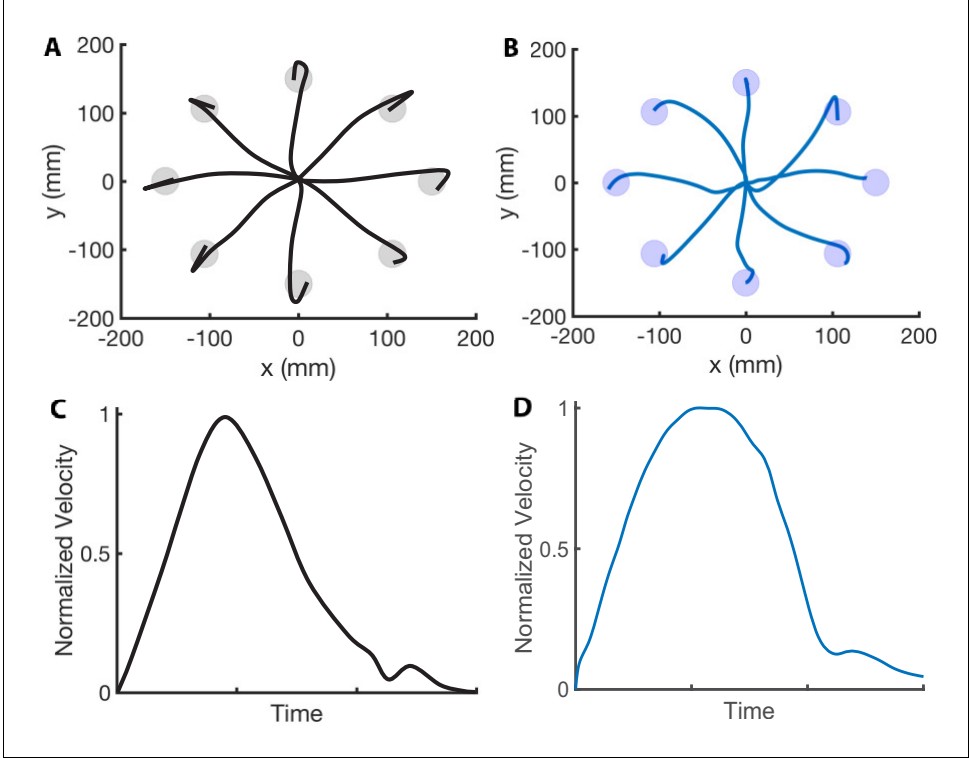

**Figure 1.** Center-out reaching. (**A** and **B**) Hand paths adapted from *Beer et al., 2000* (**A**) compared to hand paths produced by our computational model (**B**) during a center-out fast reaching task with targets placed 150 mm away from the center position. (**C** and **D**) Mean velocity profile in a reaching task adapted from *Soechting, 1984* (**C**) compared to the velocity profile produced by our computational model (**D**). We reproduce the bell-shaped velocity profile, including the smaller velocity peak at the end of the movement.

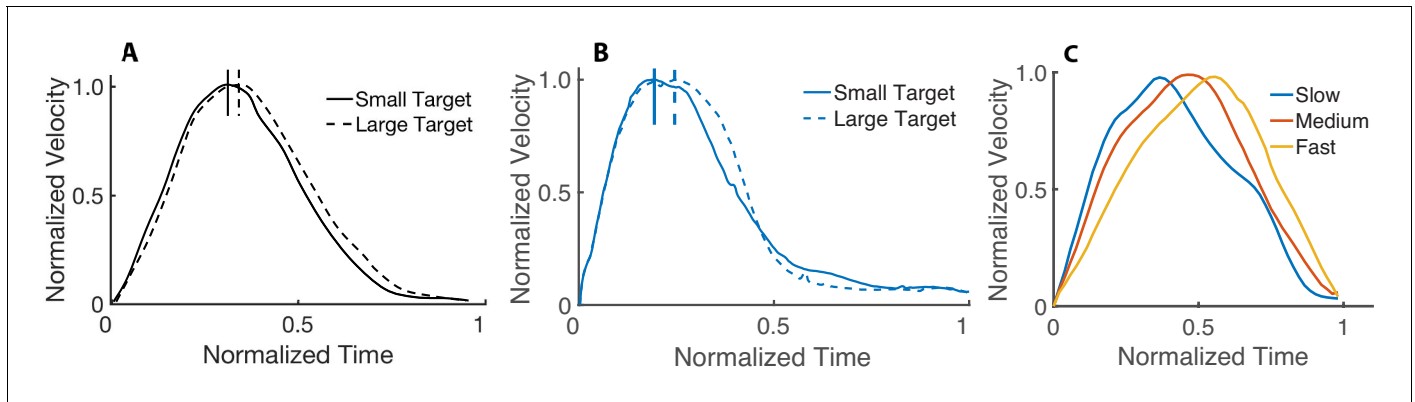

**Figure 2.** Velocity profile based on target size and speed. (**A**) Effect of the target size in reaching movements. The dashed curve is the average velocity profile for the larger target, while the solid curve is for the smallest target (adapted from *Soechting, 1984*). (**B**) Average velocity profile of 15 runs from our simulations to a small and large target. All the curves are normalized to their peak velocity. (**C**) Normalized velocity profiles (with normalized time) for fast, medium, and slow movements. Our results are consistent with the observation (e.g., *Nagasaki, 1989*) that slower movements tend to have an earlier peak velocity than faster movements.

The online version of this article includes the following figure supplement(s) for figure 2:

**Figure supplement 1.** Asymmetry of the velocity profile based on speed with a realistic biomechanical model.
**Figure supplement 2.** Asymmetry of the velocity profile based on speed with a torque-driven model.
**Figure supplement 3.** Data collection.
**Figure supplement 4.** Movement comparison.

In addition to the features reported in motor control studies, we collected three-dimensional reaching data (see the section Joint kinematics data). We compare the movements performed by a typical subject (*Figure 2—figure supplement 3*) and our computational model in *Figure 2—figure supplement 4A* and in *Video 1*. Qualitatively, we observe that the movements resemble each other, with some discrepancies such as excessive or insufficient elbow flexion. In *Figure 2—figure supplement 4B*, we compare the sagittal plane joint angles predicted by the model with our experimental data when subjects are asked to reach a shoulder flexion 180° pose with the elbow extended. The root mean square errors are 16.8° and 7° for the elbow and shoulder flexion angles.

## Speed-accuracy tradeoff and Fitts' law

Next, we solve trajectory optimization problems to targets of different sizes to study the speed-accuracy tradeoff. We keep the distance fixed to **A** = 0.15 m and vary the target width from 0.16 m to 0.0141 m, which corresponds to an index of difficulty from 0.9 to 4.41. After determining the MD for different widths, we perform a least-squares fit to determine parameters $a$ and $b$ in Fitts' law, obtaining values of 0.0234 and 0.0550, respectively. These values are within the range of values $a \in [0.0047, 0.5239]$ and $b \in [0.0393, 0.1987]$ determined from the experimental data in *Goldberg et al., 2014*, where subjects reach with a mouse to targets of various distances and widths on a computer screen.

In *Figure 3C*, we show that there is good agreement between the model's predicted MD (averaged over 10 runs) and Fitts' law ($R^2 = 0.974$). Note that the MD becomes more variable when the target size decreases as the optimization becomes more likely to fall in local minima. This feature is also present in the experimental data (*Figure 3A and B*), although it is not predicted by previous models (*Harris and Wolpert, 1998*). Although the data in *Goldberg et al., 2014* involves different experimental conditions than those of the simulated data of our center-out reach task, we expect some correlation when the index of difficulty is matched. This was verified as the correlation coefficient between our model's mean predicted MD (as it varies with the index of difficulty) and the mean of the experimental data (*Goldberg et al., 2014*; *Figure 3A*) is 0.959 (p<0.001). The correlation coefficient between the standard deviations of our model and the experimental data is 0.70 (p=0.0167). We perform the same experiment with a torque-driven model (*Figure 3—figure supplement 1*). There is again good agreement between the model's predicted mean MD (averaged over 10 runs) and Fitts' law ($R^2 = 0.88$). However, the correlations between the predicted means and standard deviations and the experimental data in *Goldberg et al., 2014* (*Figure 3A*) are weak ($R^2 = 0.17$ and $R^2 = 0.39$, respectively) and not statistically significant.

We next evaluate the sensitivity of the results with respect to the trajectory optimizer. We repeat the experiments with a less effective trajectory optimizer by limiting the number of samples and iterations to 20 and 40 (see the section Trajectory optimization), respectively. This translates to using about 12 times fewer samples during the optimization, so the computation time is about 12 times shorter. In *Figure 3—figure supplement 2*, we see that model's predicted MD means are still in close agreement with Fitts' law ($R^2 = 0.969$). The correlation coefficients between the predicted means and standard deviations with experimental data (*Figure 3A*) are significant: 0.82 (p=0.007) and 0.83 (p=0.006). Our model does not include reaction time; therefore, as expected, it predicts faster movements than those that are observed experimentally. However, the relative MDs (i.e., how the duration varies with the index of difficulty) in our model and the experimental data are still in close agreement. For example, the mean relative MD difference between an index of difficulty of 1.5 and 3.5 is about 1 s in both our model (*Figure 3—figure supplement 2*) and the experimental data (*Figure 3A*). Our cost function (*Equation 1*) includes a term to minimize the sum of muscle activations squared, which impacts the speed of reaching movements (but

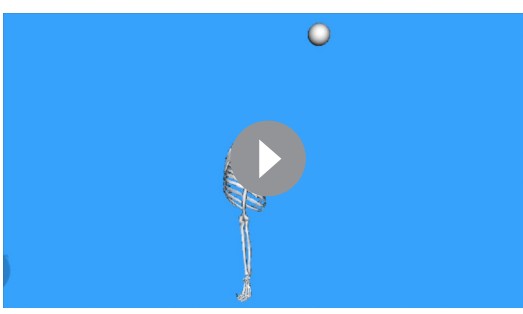

**Video 1.** Comparison of reaching movements between a representative subject (model without muscles) and our computational model (model with muscles).
https://elifesciences.org/articles/57021#video1

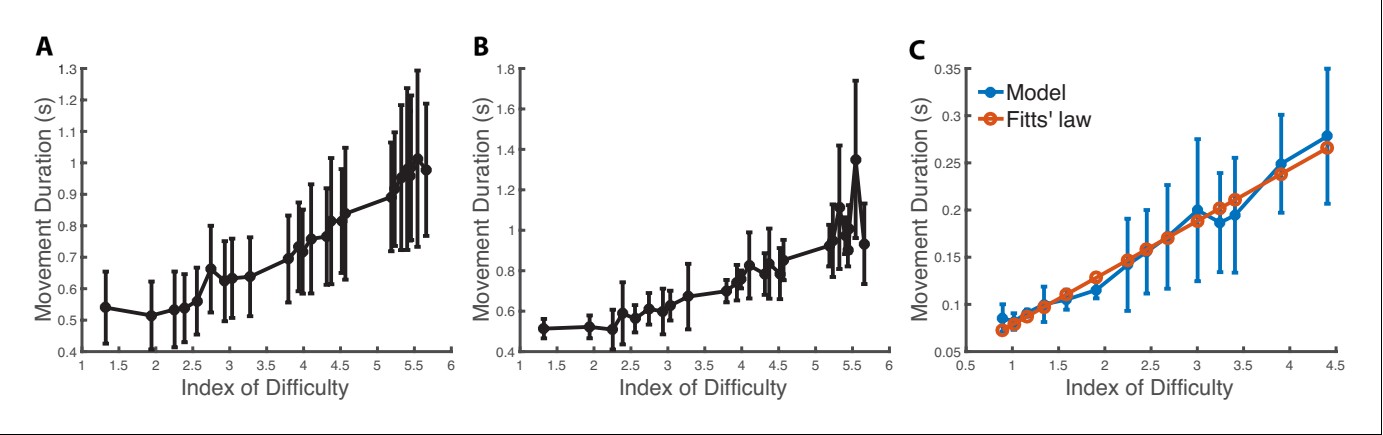

**Figure 3.** Speed-accuracy tradeoff. (**A**) Movement duration (MD) depending on the index of difficulty from the dataset in *Goldberg et al., 2014*. The mean and standard deviation are computed from 334 trials and 46 subjects (statistics computed across subjects). The experiment consists in subjects reaching with a mouse to targets of various distances and widths on a computer screen. (**B**) MD depending on the index of difficulty for a representative subject from *Goldberg et al., 2014* with statistics computed from seven trials to each target. (**C**) Predicted MD to a target in a center-out reaching task when varying the target size. The x-axis is the index of difficulty of the reaching movement, which varies from 0.9 (width: 0.16 m) to 4.41 (width: 0.014 m). The vertical bars are one standard deviation from the mean. We see that the model's mean predictions are in close agreement with Fitts' law ($R^2 = 0.974$).

The online version of this article includes the following figure supplement(s) for figure 3:

**Figure supplement 1.** Speed-accuracy tradeoff with a torque-driven model.

**Figure supplement 2.** Impact of the optimizer on the speed-accuracy tradeoff.

**Figure supplement 3.** Impact of the muscle activations term on the speed-accuracy tradeoff.

**Figure supplement 4.** Impact of the force–length relationship in muscles.

**Figure supplement 5.** Impact of a simplified model with fewer muscles.

**Figure supplement 6.** Simulated hand paths.

unlike signal-dependent noise, it does not predict different speeds based on target size). We performed an additional experiment (*Figure 3—figure supplement 3*) to show that Fitts' law emerges even without this term in the cost function. We also tested the sensitivity of our results to the biomechanical model: we performed the experiments with a model with muscles having unrealistic active force–length relationships (i.e., where the force is constant with respect to the muscle length; *Figure 3—figure supplement 4*) and a model missing important muscles (e.g., deltoid anterior, deltoid posterior, infraspinatus, and teres minor; *Figure 3—figure supplement 5*). We found that these models produced MDs that are consistent with Fitts' law, but the predictions were not correlated with the experimental data in *Goldberg et al., 2014* (*Table 1*).

## Motor cortical activity

Given the prior results regarding the potentially important role of preparatory neural activity during motor control (e.g., *Afshar et al., 2011*; *Churchland et al., 2012*; *Ames et al., 2014*; *Shenoy et al., 2013*; *Vyas et al., 2020a*), we examine the variability in the preparatory neural state during reaching movements to a small (0.75 cm) and a large (2 cm) target. We measure neural activity from premotor and primary motor cortex of a Rhesus monkey (192 simultaneously recorded channels; 96 from each brain region; see the section Neural recordings) as he makes reaching arm movements (*Figure 4A and B*). Behaviorally, we find that reaches to smaller targets are slower than those to larger targets, as predicted by Fitts' law (*Figure 4C*). The MDs to smaller targets are more variable than to larger targets (i.e., a standard deviation of 0.257 s compared to 0.234 s; p=0.0262).

Neurally, we find that reaches to smaller targets exhibit less variability in the population-level motor cortical preparatory neural state than reaches to larger targets (*Figure 4D*). Intersecting this with findings that nearly half of movement variability can be attributed to preparatory variability for a highly practiced task (*Churchland et al., 2006a*), we find that the contribution of preparatory variability to movement variability is larger for smaller targets than larger targets (*Figure 4E*). Thus,

**Table 1.** Impact of the biomechanical model.

We summarize how our results vary based on the biomechanical model. We tested six different models: a torque-driven model, a musculoskeletal model missing 14 important muscles (e.g., deltoid anterior, deltoid posterior, infraspinatus, teres minor, etc.), a musculoskeletal model with unrealistic muscle force–length relationships, a musculoskeletal model with an unrealistic cost function, a realistic musculoskeletal model with an effective optimizer, and a realistic musculoskeletal with a less effective optimizer. The results for the model 'without planning variability' are taken from the literature.

| | Fitts' law | Behavior results correlated with experimental data (MD means and standard deviations) |
|---|---|---|
| Torque or musculoskeletal model with signal-dependent noise and without planning variability | Yes | No |
| Torque model | Yes | No |
| Musculoskeletal model with missing muscles | Yes | No |
| Musculoskeletal model with unrealistic force–length relationships | Yes | No |
| Musculoskeletal model with an unrealistic cost function (i.e., no muscle activations term) | Yes | No |
| Musculoskeletal model with an effective optimizer | Yes | Yes |
| Musculoskeletal model with a less effective optimizer | Yes | Yes |

these results establish an association (albeit correlative) between Fitts' law and variability in neural activity during motor preparation.

## Discussion

In this study, we investigate a fundamental phenomenon underlying goal-directed motor control. Fitts' law formalizes the notion that movement time is a function of both the distance to a target and the size of that target. Taken together with a widely held theory initially proposed by *Harris and Wolpert, 1998*, Fitts' law is thought to arise from the fact that signal-dependent noise governs a person's ability to move with greater accuracy toward smaller targets by reducing movement speed. Signal-dependent noise or 'motor noise' has been shown across a variety of experiments, where noise scales in proportion to the size of the motor command being generated (e.g., *Jones et al., 2002*; *Matthews, 1996*). Thus, to a first-order approximation, signal-dependent noise is a sufficient theory to explain Fitts' law. Here, we re-examine the assumption that Fitts' law emerges primarily as a *consequence* of signal-dependent noise.

We develop a computational model of three-dimensional upper extremity movements that is more biomechanically realistic than previous work. Our model replicates kinematic features reported across several motor control studies. When solving the optimal control problem, we find that some of our simulations reproduce the empirical observation that arm movement paths are nearly straight and gently curved near the targets for fast reaches. The corresponding velocity profiles are bell-shaped and have a secondary, smaller peak near the target. These features are usually interpreted as discrete, open-loop corrections to acquire the target (*Doeringer and Hogan, 1998*). While our results are consistent with this interpretation, our analyses also show that these features are present in some local minima when solving the optimal control problem. Out of 20 trials during a reaching task, we consistently note that the second or third best solution (measured with respect to the cost; *Equation 1*) has these features, while the best solution consists of a nearly straight trajectory. The curve and the secondary velocity peak at the end of the movement are a consequence of the hand reaching the target with a high velocity; the final, gentle curve, brings the hand velocity toward zero, minimizes 'effort', and keeps the hand inside the target.

*Bullock and Grossberg, 1988* proposed that the speed-dependent asymmetry in the velocity profile arises due to neural dynamics and that models based on optimal control always predict symmetric velocity profiles. Our work demonstrates that asymmetric velocity profiles in reaching movements arise from optimal control methods with realistic biomechanical models, without relying on other modeling assumptions such as time-varying jerk constraints (*Nagasaki, 1989*). This feature did not emerge in earlier work using simplified biomechanical models (*Guigon et al., 2007*; *Harris and*

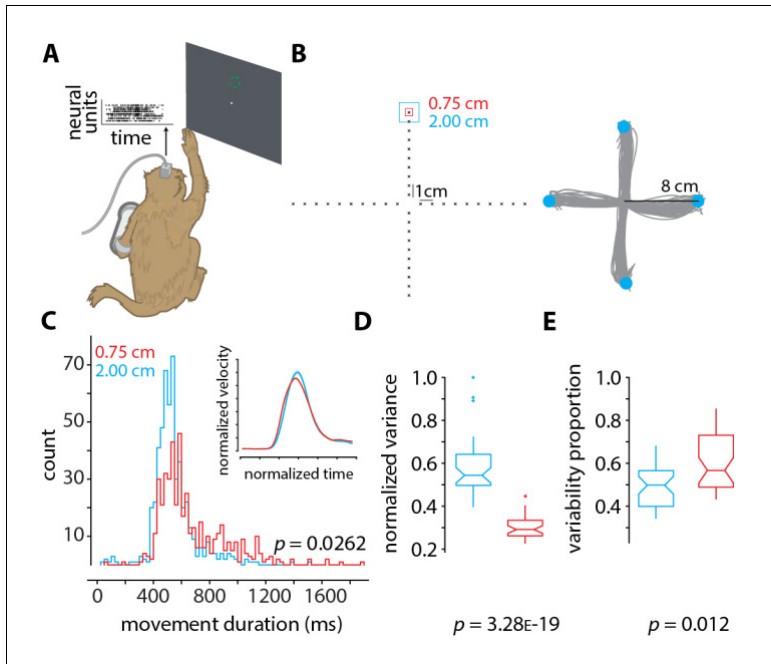

**Figure 4.** Neural data and analysis. (**A**) Rhesus monkey reaches in 3D space using his right arm, while neural activity from 192 channels are recorded from two Utah arrays surgically implanted into contralateral dorsal premotor and primary motor cortex. The monkey's arm position in space controlled the velocity of the cursor on the screen (methods described in *Even-Chen et al., 2019*). (**B**) (Left) the layout of the 40 targets to which the monkey reaches. For each of the 40 targets, two conditions are presented in a block-wise format. In one condition (block) the target has an acceptance window of size 2 cm (cyan), and in the other the acceptance window is of size 0.75 cm (red). Within each block, the targets appear randomly. Each block contains 400 trials (10 trials per target), and each experimental session contains about 10 blocks, 5 of each condition. (Right) representative arm reaches made by the monkey for four targets 8 cm away from the center with an acceptance window size of 2 cm. (**C**) Histograms of movement duration (MD) for the large (2 cm) and small (0.75 cm) targets (inset shows the normalized velocity profiles). The observed difference in variance is statistically significant (p=0.0262 via two-sample F-test). The MDs to the large and small targets are 461 ± 234 ms and 661 ± 258 ms. (**D**) Normalized variance of the preparatory neural state for the small (0.75 cm targets, red) and large (2.00 cm targets, cyan) targets. The variance in the preparatory state is computed by first taking trial-averaged firing rates for the last 200 ms of preparation before the go-cue, performing principal components analysis, and finding the volume of the error ellipse in 3D space, which captures at least 80% of the total variance (similar to as described in *Vyas et al., 2018* and *Even-Chen et al., 2019*). Note that we do not spike sort or assign spikes to individual neurons (*Wood et al., 2004*). We instead use threshold crossings (*Trautmann et al., 2019*). (**E**) Proportion of movement variability explained by preparatory variability (as described in *Churchland et al., 2006a*), for the small (0.75 cm targets, red) and large (2.00 cm targets, cyan) targets. In **D** and **E** the p-values are computed from one-way ANOVA.

*Wolpert, 1998*). The upper extremity must decelerate when approaching the target to achieve zero velocity. The elbow flexors undergoing an eccentric contraction have a larger force generating capacity at higher speeds due to the force–velocity relationship of muscle (*Millard et al., 2013*). Our hypothesis is that this explains why faster movements have a later peak velocity. We have verified that this property does not arise in a torque-driven model (*Figure 2—figure supplement 2*), which supports the hypothesis.

Reaching involves feedback control, which we did not explicitly model as this is outside the scope of our study (we are not aware of any validated feedback models on realistic upper-extremity musculoskeletal models in the literature). We do not expect that explicitly modeling feedback would influence the results. Feedback would ensure that the hand successfully reaches the target in the presence of noise, but planning variability would still impact the MD.

During motor learning, there is a shift in performance that allows movements to be performed faster and more accurately. Previous studies hypothesize that this shift in performance is a result of

improvements in the signal-to-noise ratio (i.e., the ability to use large control signals without large motor noise). This is perhaps possible due to expanded neural representations and/or the fine-tuning of individual neurons (*Krakauer and Mazzoni, 2011*; *Shmuelof et al., 2012*). Indeed, there is evidence of an improved signal-to-noise ratio during motor skill learning in rodents (*Kargo and Nitz, 2004*), but their mechanism proposes a reduction in signal-independent noise and pre-burst firing rates. If signal-dependent motor noise alone accounts for the speed-accuracy tradeoff, the mechanism that explains the shift in performance due to practice is not yet fully understood. Our analyses reveal that more stringent task constraints reduce the space of possible control solutions, thereby removing effective solutions from the search landscape (*Figure 5*). The controller (i.e., the central nervous system) is unable to find the global optimum solution for every task. Thus, for challenging tasks, the patterns of neural activity found during initial optimization (i.e., learning) correspond to ineffective solutions, leading to suboptimal behavior, e.g., reaching slower to smaller targets, consistent with Fitts' law. Learning can force the motor system to explore different solutions, and if better ones are found, the internal models are updated accordingly resulting in behavioral improvements (*Haith and Krakauer, 2018*). Hence, our theory does not rely on assumptions regarding improvements in signal-to-noise ratio to explain the shift in performance with practice. A testable prediction of our theory is that the neural variability during motor planning should progressively decrease as the shift in performance occurs during practice (and as the central nervous system converges to better motor plans).

While our theory does not account for signal-dependent noise, the broader mechanism for motor control likely engages both the mechanism proposed here and certain aspects of signal-dependent noise. We exclude signal-dependent noise in our model to demonstrate that it is not necessary to explain the speed-accuracy tradeoff. Adding signal-dependent noise with the same cost function and optimization procedure (that includes motor planning variability) would also yield Fitts' law-like solutions, but the signal-to-noise ratio would limit how fast the reach to the target can be achieved. One difficulty with modeling signal-dependent noise is that there is very little experimental data on the signal-to-noise ratio and how it changes with practice – if it changes at all. Future work exploring tasks where subjects learn a task across sessions could be quite useful in making more progress toward answering this question. We posit that a realistic biomechanical model, as described here,

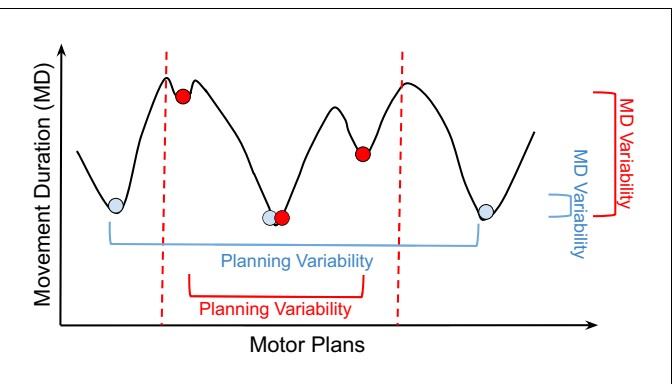

**Figure 5.** Optimization landscape. Cartoon of how different motor plans can generate movements (i.e., driven by muscle excitation patterns) with different durations. The blue and red circles are examples of control solutions for the large and small targets, respectively. The dashed lines illustrate the constraints in the optimization landscape for the smaller target. From the analysis in *Figure 4C and D*, for reaches to a small target (red), movement planning variability decreases (x-axis), but movement duration (MD) variability increases (y-axis). For reaches to a larger target (blue), movement planning variability increases, but MD variability decreases. According to our theory, reaches to larger targets are faster because of the presence of good control solutions in the optimization landscape. We empirically observe that these good solutions have MDs that are only slightly larger than the global minimum solution (i.e., the fastest reach to the target). A bad control solution corresponds to a significantly slower reach. On average, reaches to the small target are slower because tighter constraints remove good solutions from the optimization landscape, which is otherwise identical to the landscape for the large target. While this decreases motor planning variability, it increases MD variability. In stochastic optimization, it is empirically found that having a large number of good local minima helps avoid bad local minima (*Goodfellow et al., 2016*, Chapter 8).

allows for more accurate estimation of the optimization landscape (i.e., how the local optima are situated with respect to the control solutions). This explains why the realistic biomechanical model produces results that are strongly correlated with prior experimental findings (*Goldberg et al., 2014*), while all of our simplified models (which includes the torque-driven, fewer muscles, unrealistic force-length relationships, and unrealistic cost function models) are not statistically correlated with prior experimental findings since they are more likely to synthesize unrealistic movements (*Table 1*).

Our experiments show that Fitts' law emerges both with a standard and a less effective optimizer. This follows from the fact that for an optimizer with a given performance (i.e., measured by the number of samples and trials that it can explore), imposing progressively more stringent accuracy constraints removes good solutions from the optimization landscape, resulting in performance degradation and thus yielding Fitts' law. Interesting questions for future work include finding the number of samples and trials in our stochastic optimizer and exploring other forms of stochastic optimization (e.g., *Lillicrap et al., 2015*) that yields solutions that best match the experimental data on the speed-accuracy tradeoff. A theory based purely on signal-dependent motor noise does not predict that more severe accuracy constraints cause more MD variability, as seen in the experimental data. From this perspective, our theory provides a more complete account of the speed-accuracy tradeoff.

If our theory is correct, then the muscle excitations, activations, and joint kinematics predicted for an ineffective solution should be consistent with what is empirically observed in naïve or 'non-expert' human participants. At the same time, our theory predicts that an effective solution would result in muscle patterns that are consistent with empirical observations of highly trained or 'experts' performing the task. Future studies can perform this stringent test of our theory by utilizing our modeling framework to find the control solutions for some subset of challenging tasks and compare them against muscle activity measured in a laboratory setting.

In addition to planning variability, limb dynamics play a role in the speed-accuracy tradeoff: one reason why the movement to a smaller target is slower than a larger target with the same center is that a longer distance is required to reach the smaller target; hence, the larger target contains both a larger set of feasible solutions and more effective solutions. The differences in durations between the fastest trials to the smallest and largest targets provide an indicator of the role of limb dynamics in the speed-accuracy tradeoff. The fastest trial approaches the limit of what is dynamically feasible given enough trials. In our computational experiments, this difference amounts to 0.04 s. The average difference from the experimental data is also 0.04 ± 0.03 s (*Supplementary file 1*). The difference between the mean durations to the two targets minus the difference between the fastest reaches (65% of the total difference in *Figure 3C*) is attributable to motor planning variability (as we have excluded the roles of limb dynamics and execution noise), which is the dominant factor behind the speed-accuracy tradeoff in our results. The proportions would likely be reversed if the target sizes were drastically different (e.g., comparing targets that were an order of magnitude apart in size).

Our analysis does not preclude the possibility that signal-dependent noise is a factor in the speed-accuracy tradeoff in addition to limb dynamics and planning variability, but it shows that it is not a necessary condition and may not be the dominant factor as previously thought. The data in *Supplementary file 1* also shows that the subjects that achieved the smallest difference in durations between the two targets on their best trials also had the largest variability (see subjects 1, 2, and 3). We hypothesize that this variability could be attributed to exploration, which allows for finding better control solutions, consistent with the experimental data. This is also consistent with recent findings that learning is associated with increases in task-relevant variability (*Wu et al., 2014*).

Turning our attention to neural mechanisms underlying motor cortical function, a current theory suggests that movement period neural activity evolves from an initial condition (termed the neural population preparatory state) set in premotor and primary motor cortex (*Shenoy et al., 2013*; *Vyas et al., 2020a*). Different movements have different initial conditions and generating initial conditions closer to the optimal initial condition results in behavioral benefits (*Afshar et al., 2011*). Furthermore, trial-by-trial behavior variability is strongly linked to neural variability during motor preparation, even for highly practiced movements (*Churchland et al., 2006a*), and motor learning is strongly linked with systematic changes to neural activity during motor preparation, where neural activity during motor preparation plays a causal role in error-driven learning (*Vyas et al., 2018*; *Vyas et al., 2020b*). Given these findings, we introduce a computational model with planning

variability and demonstrate its ability to accurately recapitulate many previous results and phenomenon, including Fitts' law, which suggests a potential role in the speed-accuracy tradeoff. If movement planning variability is a dominant factor underlying the speed-accuracy tradeoff, then the proportion of MD variability explained by preparatory variability should increase with reaches to smaller targets (otherwise motor planning does not contribute to the speed-accuracy tradeoff). We have experimentally verified this prediction (*Figure 4E*). Our results show that nearly 60% of the MD variability is attributed to planning variability when reaching to the smaller target (and that the MD standard deviation is large relative to the mean with a coefficient of variation of 0.4). When combined with the observation that MD variability increases for smaller targets, this indicates that motor planning variability impacts the scalar parameters found in Fitts' law (see the section Introduction). Our experiments here further show that reaches to smaller targets are slower and less variable in preparatory activity than reaches to larger targets (*Figure 4D*), while the MDs are more variable in smaller targets than in larger targets. If the speed-accuracy tradeoff was solely caused by execution noise, then we would expect MD variability to decrease when planning variability decreases. These results, however, are consistent with our theory (*Figure 5*), which suggests that some of the effective control solutions (i.e., motor plans for fast reaches) have to be discarded when reaching to smaller targets (i.e., variability in preparatory activity decreases, while MD variability increases). We can estimate the relative contribution of planning variability to the speed-accuracy tradeoff as follows. In *Figure 4C*, the mean difference in the MD to the large and small target is 200 ms, and the standard deviation of the MD to the small target is 258 ms. Given that planning variability explains 60% of the MD variability (*Figure 4E*), and if we assume a normal distribution around the mean MD, then motor planning can cause the movement to the small target to be 155 ms faster (i.e., one standard deviation from the mean). This suggests that planning variability can account for more than 75% of the MD difference when reaching to the large and small target (i.e., $0.6 \times 258/200$). We note that this is an initial *theory*, and future studies will need to evaluate its range of applications under additional experimental conditions.

Our results here (*Figure 4D*), and those by Churchland and colleagues (*Churchland et al., 2006a*; *Churchland et al., 2006b*; *Churchland et al., 2006c*), point to preparatory activity as also being a central source of movement variability, rather than execution noise alone being the key factor. Our work presumes that the movement variability resulting from motor planning variability can be modeled with the solutions found from stochastic optimization (but we do not claim that a stochastic optimization is actually being literally performed by the brain during motor planning). In other words, during stochastic optimization, many possible candidate motor plans will be explored before converging to a particular motor plan. Our account of Fitts' law does not depend on the optimization time or of the candidate motor plans explored before convergence. Rather, our results show that, at convergence (i.e., the solutions from stochastic optimization), when reaching to smaller targets, the motor plans tend to be less effective (i.e., slower reaches) because of task constraints (*Figure 5*). When comparing the reaches to a large and a small target, we have empirically verified that the reaches to the large target that also do not reach the small target are on average faster than the reaches to the small target. Our computational model implements motor planning stochasticity with covariance matrix adaptation evolution strategy (CMA-ES), which operates by applying Gaussian noise on an initial motor plan. It is unknown how motor planning stochasticity arises at the individual neural and neural population level, but it could be as simple as noise on an initial motor plan (e.g., due to variability in the inputs from other brain regions [*Vyas et al., 2020b*]). We have approximated the trial-to-trial variability in the preparatory state in our model via sampling from a Gaussian distribution. Although we cannot describe the exact neurophysiological mechanism during preparation that produces differential effects on variability when reaching to targets of different size, our contribution lies in discovering this relationship. In summary, we know from neurophysiological findings that a large proportion of the behavioral variability occurring in highly practiced tasks can be attributed to motor planning variability (*Churchland et al., 2006a*). When implementing planning variability in a computational model, we obtain MDs that are consistent with Fitts' law and experimental findings on MD variability (*Figure 3C*). This indicates a potential role for planning variability in giving rise to the speed-accuracy tradeoff, which is supported by further neurophysiological evidence introduced here (*Figure 4D* and *Figure 4E*).

It is important to note a few important caveats and considerations regarding our theory. First, we do not have causal evidence linking neural variability during preparation and Fitts' law (although we

have a causal link between variability during preparation and Fitts' law in our computational model). Second, we cannot assert that stochasticity in the model's motor plan is a perfect proxy for variance in the motor cortical preparatory state, but our working hypothesis is that there is enough alignment for our theory to hold given that our simulations match well with experimental data (see the section Speed-accuracy tradeoff and Fitts' law); future work will need to be done to explore the extent of this alignment. Finally, while the preparatory state does not encode the whole reach trajectory (as the model's motor plan does), it does encode important parameters of the upcoming reach, such as distance, direction, and speed (*Messier and Kalaska, 2000*; *Churchland et al., 2006b*; *Even-Chen et al., 2019*; *Vyas et al., 2020a*). One tantalizing interpretation of our results is that Fitts' law is tightly linked to movement planning (which includes many of the aspects of the preparatory activity analyzed here) and not just movement execution. The difficulty associated with finding optimal solutions for challenging tasks could then be akin to finding the optimal preparatory neural states (as well as learning the appropriate movement period dynamics), where task constraints govern the optimization landscape, as suggested by our theory. This provides a concrete avenue for future studies to further investigate the relative contribution of signal-dependent noise and motor planning in giving rise to the speed-accuracy tradeoff.

## Materials and methods

### Modeling and simulation

Our computational model uses the upper extremity biomechanical model introduced by *Saul et al., 2015* and available at https://simtk.org/projects/upexdyn. The model consists of 47 Hill-type muscle-tendon actuators (*Zajac, 1989*) with parameters and paths derived from experimental and anatomical studies. The skeletal model has five degrees-of-freedom representing the shoulder (elevation plane, elevation angle, and shoulder rotation), the elbow (flexion/extension), and the forearm (pronation/supination). We have locked the wrist and finger joints. The simulation was performed with OpenSim (*Delp and Anderson, 2007*) and the Simbody physics engine (*Sherman et al., 2011*) with a semi-explicit Euler integrator (accuracy 1e−2).

### Trajectory optimization

Numerical methods for solving trajectory optimization problems is an ongoing research topic (e.g., *Al Borno and Lasa, 2012*, *Posa and Tedrake, 2013*, *Todorov and Li, 2005*). Our work adapts the method of Al Borno et al. 2013 to optimize the movement of musculoskeletal systems. We optimize for a time-indexed cubic B-spline that provides the muscle excitations to produce movement. The optimization is performed with CMA-ES (*Hansen, 2006*). The free variables are the spline knots muscle excitations values at every 0.1 s interval in the movement. We use 500 iterations and a population size of 20 in CMA-ES, which takes about 3 hr of computation time (after parallelization) with two Intel Xeon CPU E5-4640 processors. We initialize the optimization with all the knots set to 0.1. Given excitations, we use a forward simulation to obtain a trajectory, which is evaluated with respect to the following cost function:

$$F = w_1 \left\| \boldsymbol{h}_{\mathrm{N}} - \boldsymbol{h}_{\mathrm{N}}^{\mathrm{d}} \right\|^2 + w_2 \delta \left( \sum_{i=1}^{\mathrm{N}} \left\| \boldsymbol{a}_i \right\|^2, \zeta \right) + w_3 \sum_{i=1}^{\mathrm{N}} l(\boldsymbol{q}_i) + w_4 \delta(\left\| \dot{\boldsymbol{q}}_{\mathrm{N}} \right\|^2, \zeta) + w_5 \rho(\boldsymbol{q}_{\mathrm{N}}) \tag{1}$$

where $\boldsymbol{h}_{\mathrm{N}}$ and $\boldsymbol{h}_{\mathrm{N}}^{\mathrm{d}}$ denote the actual and desired position of the center of the hand at the last time step $\mathrm{N}$ of the trajectory, $\boldsymbol{a}_i$ denotes the vector of muscle activations at time step $i$, $l(\boldsymbol{q}_i)$ is a penalty on joint limit violations given pose $\boldsymbol{q}_i$, $\dot{\boldsymbol{q}}_{\mathrm{N}}$ denotes the joint velocities at the last time step, $\rho(\boldsymbol{q}_{\mathrm{N}})$ is a term to encourage pronation of the forearm at the last time step, and $\| \cdot \|$ denotes the two-norm. We now give the details of the terms $l(\boldsymbol{q}_i)$ and $\rho(\boldsymbol{q}_{\mathrm{N}})$. For a joint value $r_i$ with limits $y$ and $u$ (denoting the lower and upper bounds for the coordinate), $l(r_i)$ denotes a quadratic penalty on joint limit violations:

$$l(r_i) = \begin{cases} 0 & \text{if} \quad y < r_i < u \\ (r_i - y)^2 & \text{if} \quad r_i < y \\ (r_i - u)^2 & \text{if} \quad r_i > u \end{cases}$$

The $l(\boldsymbol{q}_i)$ term is the sum of the penalties for each joint. The term $\rho(\boldsymbol{q}_N)$ is defined as:

$$\rho(\boldsymbol{q}_N) = (r_{\text{forearm}} - \phi)^2$$

where $r_{\text{forearm}}$ denotes the coordinate of the forearm and $\phi$ is the forearm value denoting pronation. When reaching, the position of the forearm is chosen based on the target object (e.g., a cup or a light bulb). Since we do not model object grasping, we chose to consistently reach with the forearm in pronation at the last time step. It is challenging to tune the weights in the cost function as objectives tend to 'fight' each other. For this reason, we introduce the $\delta(\cdot, \cdot)$ function:

$$\delta(\text{x}, \zeta) = \begin{cases} \text{x if } \left\| \boldsymbol{h}_N - \boldsymbol{h}_N^{\text{d}} \right\|^2 < \zeta \\ C \text{ otherwise} \end{cases}$$

where we set the scalars $\zeta$ to 0.025 and $C$ to $10^6$ (i.e., $C \gg \text{x}$). This ensures that the optimization prioritizes finding a trajectory with the hand reaching close enough to the target (i.e., within $\sqrt{\zeta}$ m) before considering other objectives, namely minimizing the sum of activations squared and achieving a final zero velocity. We tuned the weights $w_1$, $w_2$, $w_3$, $w_4$, and $w_5$ to 5, 1e−3, 1e−5, 0.15, and 0.5. The cost function encourages the hand to be as close as possible to a desired target with a small velocity and the forearm in pronation, while minimizing muscle activations and joint limit violations throughout the movement. When the objective is to reach a target anywhere within a radius $\kappa$, then we simply set $w_1$ to zero when $\| \boldsymbol{h}_N - \boldsymbol{h}_N^{\text{d}} \| < \kappa$.

The movement duration T is chosen based on experimental data or through optimization to study the speed-accuracy tradeoff. For the latter, we encourage the movement to reach the target as soon as possible by including the objective $\phi(\boldsymbol{h}_i)$ in the cost function (*Equation 1*). We set $\phi(\boldsymbol{h}_i)$ to 0 if the target is reached at time step $i$; otherwise, we set it to 1 (i.e., a penalty at each time step where the hand position is outside the target). We perform the trajectory optimization with a fixed temporal horizon and the MD is determined when the hand position is within the target. We performed experiments to verify that prescribing the muscle excitations at every 0.05 s interval (instead of 0.1 s interval) does not have a significant impact on the computed MDs. *Haith et al., 2012* argue that a hyperbolic cost of time objective term better accounts for the durations of saccades, but this remains a topic of investigation for other body movements. After setting the duration of the movement T for the integration, the number of time steps N in the trajectory is chosen to achieve the desired accuracy in Simbody. For the experiments with our torque-driven model, we optimize for the torques, which we limit to ±50 nm, instead of muscle excitations.

## Joint kinematics data

To validate our computational model, we collected three-dimensional upper extremity movement data. Right-handed subjects (n = 4, age = 27 ± 3, mean ± std, two males, two females) gave written informed consent approved by the Stanford University Institutional Review Board. The subjects are asked to reach for desired final end poses (e.g., with shoulder abduction 90˚, shoulder flexion 90˚ and 180˚, etc.) from different initial positions. In *Figure 2—figure supplement 3*, we show an example of an initial (A) and final position (B). Subjects are instructed to reach the target pose at a comfortable speed, without moving their trunk or feet. Upper extremity kinematics are measured with inertial motion capture (*Roetenberg et al., 2009*). MD and the final hand position (computed from forward kinematics) are then used for motion synthesis with trajectory optimization, using the final hand position for the target position $\boldsymbol{h}_N^{\text{d}}$.

## Fast reaching task

We collected experimental data from five right-handed subjects (n = 5, age = 28 ± 4.9, mean ± std, four males, one female) that gave informed consent approved by the Stanford University Institutional Review Board. The subjects performed reaches to a target, always starting from the same initial pose with the hand on the table. Subjects were instructed to reach the target as fast as possible.

The duration of the reach is measured from an accelerometer attached on their hand (Xsens MTw Awinda with a 1000 Hz sampling frequency). Subjects performed 10 reaches to a large square target (8 cm width) and to a small one (2 cm width). The movement amplitude was set to 15 cm. The data is presented in *Supplementary file 1*.

### Neural recordings

Recordings were made from the dorsal aspect of the premotor cortex (PMd) and the primary motor cortex (M1) of one male adult monkey (*Macaca mulatta*) who was 15 years old and weighed 16 kg at the time of these experiments. The monkey performed an instructed-delay cursor task (see *Vyas et al., 2018* for the task design and standard parameters). The monkey had two chronic 96-electrode arrays (1 mm electrodes, 1.0 mm long, spaced 400 μm apart, Blackrock Microsystems Inc, Salt Lake City, UT), one implanted in PMd and one implanted in gyral M1. The arrays were implanted 7 years prior to these experiments. Voltage signals were band-pass filtered from each electrode channel (250 Hz to 7.5 KHz). The signals were processed to detect multi-unit 'threshold crossing' spikes. Spikes were detected any time the voltage crossed below a threshold of $-4.5$ times the root-mean-square voltage.

The standard methods for an instructed-delay reaching task, including the reward structure, arm-tracking system, inter-trial-intervals, etc. have been previously described (*Vyas et al., 2018*). The 40 targets that the animals made reaches to were arranged in '+' configuration; however, only one target (chosen pseudo-randomly) was presented during each trial. The target presentations were chosen to ensure that the animal could not predict the next target and that the animal made an equal number of reaches to each target. For each of the 40 targets, two conditions were presented in a block-wise format. In one condition (block) the target had an acceptance window of size 2 cm (cyan in *Figure 4*), and in the other the acceptance window was of size 0.75 cm (red in *Figure 4*). That is to say, the animal had to bring the cursor he was controlling within the acceptance window and hold it there for 500 ms in order for the trial to succeed and for him to receive his liquid reward. A trial was failed if (1) the animal did not complete the trial in the allotted time (maximum of 2 s), (2) the animal did not hold the cursor in the center for the allotted period of time, or (3) the animal failed to initiate the trial within 500 ms. Each block contained 400 trials (10 trials per condition) and each experimental session contained about 10 blocks, 5 of each condition.

The analysis presented in *Figure 4D* was performed following the standard methods described in *Vyas et al., 2018*; *Even-Chen et al., 2019*; *Vyas et al., 2020b*. The analysis presented in *Figure 4E* was performed following the standard methods described in *Churchland et al., 2006a*.

## Acknowledgements

We thank Adrian Haith for very helpful comments on the manuscript. We thank Mackenzie Risch and Michelle Wechsler for expert surgical assistance and veterinary care. We thank W L Gore Inc for donating Preclude artificial dura used as part of the chronic electrode array implantation procedure. We thank Dr. Stephen I Ryu for surgical assistance. SV was supported by an NIH F31 Ruth L Kirsch-stein National Research Service Award 5F31NS103409-02, an NSF Graduate Research Fellowship, and a Ric Weiland Stanford Graduate Fellowship. KVS was supported by the following awards: National Institutes of Health (NIH) National Institute of Neurological Disorders and Stroke (NINDS) Transformative Research Award R01NS076460, NIH National Institute of Mental Health Grant (NIMH) Transformative Research Award R01MH09964703, NIH Director's Pioneer Award8DP1HD075623, Defense Advanced Research Projects Agency (DARPA) Biological Technology Office (BTO) 'REPAIR'' award N66001-10-C-2010, DARPABTO "NeuroFAST'' award W911NF-14-2-0013, the Simons Foundation Collaboration on the Global Brain awards 325380 and 543045, ONR and the Howard Hughes Medical Institute. SLD was supported by the Mobilize Center, a National Institutes of Health Big Data to Knowledge (BD2K) Center of Excellence through Grant U54EB020405.

## Additional information

### Funding

| Funder | Grant reference number | Author |
|---|---|---|
| National Institutes of Health | U54EB020405 | Scott L Delp |
| National Institute of Neurological Disorders and Stroke | R01NS076460 | Krishna V Shenoy |
| National Institute of Mental Health | R01MH09964703 | Krishna V Shenoy |
| Defense Advanced Research Projects Agency | N66001-10-C-2010 | Krishna V Shenoy |
| National Institutes of Health | 8DP1HD075623 | Krishna V Shenoy |
| Simons Foundation | 325380 and 543045 | Krishna V Shenoy |
| Defense Advanced Research Projects Agency | W911NF-14-2-0013 | Krishna V Shenoy |
| Howard Hughes Medical Institute | | Krishna V Shenoy |
| National Institutes of Health | 5F31NS103409-02 | Saurabh Vyas |
| National Science Foundation | Graduate Fellowship | Saurabh Vyas |
| Stanford University | Ric Weiland Stanford Graduate Fellowship | Saurabh Vyas |

The funders had no role in study design, data collection and interpretation, or the decision to submit the work for publication.

### Author contributions

Mazen Al Borno, Conceptualization, Software, Formal analysis, Investigation, Writing - original draft; Saurabh Vyas, Formal analysis, Investigation, Writing - original draft; Krishna V Shenoy, Scott L Delp, Supervision, Funding acquisition, Writing - review and editing

### Author ORCIDs

Mazen Al Borno (iD) https://orcid.org/0000-0003-2208-9934

### Ethics

Human subjects: Subjects gave written informed consent, and consent to publish, approved by the Stanford University Institutional Review Board (42787). The guidelines followed are specified in the Human Research Protection Program (HRPP Stanford University).
Animal experimentation: All surgical and animal care procedures were performed in accordance with National Institutes of Health guidelines and were approved by the Stanford University Institutional Animal Care and Use Committee (8856).

### Decision letter and Author response

Decision letter https://doi.org/10.7554/eLife.57021.sa1
Author response https://doi.org/10.7554/eLife.57021.sa2

## Additional files

### Supplementary files

• Supplementary file 1. Fast reaching experimental data. For five subjects, we present the mean, standard deviation, minimum, and maximum of the movement durations in 10 reaches to a large (L) and small (S) square targets (see the section Fast reaching task). The target widths are 8 cm and 2

cm, and the reach amplitudes are 15 cm. The data are available at https://simtk.org/projects/ue-reaching/.

• Transparent reporting form

### Data availability

The source code for the computer simulations and our data are available at https://simtk.org/projects/ue-reaching. Users must first create a free account (https://simtk.org/account/register.php) before they can download the datasets from the site.

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
