## [Decision Letter]

**Acceptance summary:**

The present modelling study and non-human primate experiments suggest a new theory for the origin of the speed-accuracy tradeoff, a key feature of motor behavior across effectors and contexts. The authors posit that, rather than signal-dependent noise, the speed-accuracy trade-off arises because of planning variability. That is, more difficult task demands (i.e. smaller targets) reduce the availability of good control solutions (i.e. fast reaches).

**Decision letter after peer review:**

Thank you for submitting your article "High-fidelity musculoskeletal modeling reveals a motor planning contribution to the speed-accuracy tradeoff" for consideration by *eLife*. Your article has been reviewed by three peer reviewers, including Andrew Pruszynski as the Reviewing Editor and Reviewer #1, and the evaluation has been overseen by Ronald Calabrese as the Senior Editor. The following individual involved in review of your submission has agreed to reveal their identity: Friedl de Groote (Reviewer #3).

The reviewers have discussed the reviews with one another and the Reviewing Editor has drafted this decision to help you prepare a revised submission.

As the editors have judged that your manuscript is of interest, but as described below that additional work is required before it is published, we would like to draw your attention to changes in our revision policy that we have made in response to COVID-19 (https://elifesciences.org/articles/57162). First, because many researchers have temporarily lost access to the labs, we will give authors as much time as they need to submit revised manuscripts. We are also offering, if you choose, to post the manuscript to bioRxiv (if it is not already there) along with this decision letter and a formal designation that the manuscript is "in revision at *eLife*". Please let us know if you would like to pursue this option. (If your work is more suitable for medRxiv, you will need to post the preprint yourself, as the mechanisms for us to do so are still in development.)

Summary:

The present paper presents a novel theory for the origin of Fitts' Law, that is, the speed-accuracy trade-off observed across movement tasks. The most prominent current theory suggests that the speed-accuracy tradeoff emerges due to the optimization of movements in the face of signal-dependent noise. The authors suggest that, under a physiologically realistic simulation of the arm, motor planning in the absence of motor noise can also give rise the speed-accuracy trade-off. The simulation predictions are coupled with behavioral data from a straightforward human reaching experiment and the simulations are linked to a potential underlying neural mechanism via single-units recordings obtained from a macaque monkey doing a similar reaching task.

All three reviewers thought that this is an interesting and provocative study that may provide an important new perspective on the origins of Fitts' Law, the implications of which would be very broad. However, all three reviewers converged on a set of common concerns about the current version of the paper that need to be addressed.

Essential revisions:

1) All the reviewers raised substantial concerns about how the authors handle physiological noise in their simulations. There were two related issues.

a) In the present version of the model, muscle activation terms are squared and the authors suggest that this serves as an approximation of metabolic power consumption. The reviewers raised the point that, in other contexts, minimizing squared muscle activations has been interpreted as minimizing motor noise (i.e. acts like the kind of signal-dependent noise term). It is important that the authors clarify how this term influences the ability to make a clean separation between the hypotheses being tested.

b) Physiological noise is an established phenomenon that is not included in the present model. The reviewers understand why the authors chose this route (both conceptually and practically) but thought it would be important to show what happens when physiological noise is included. In this case, would the same optimization procedure with the same cost function converge similarly on a Fitts law-like solution? How would the optimized control signals in this case differ from the ones they found presently?

2) All the reviewers raised the substantial concern that the present work does not establish a meaningful link between the optimal control model being proposed – and its reasons for showing Fitts' Law – and a sensible neurophysiological analog. That is, what neurally feasible computational mechanism occurs during movement preparation that produces greater variability for small targets. Given the present model algorithm does not seem feasible, what to the authors have in mind? At the very least, the authors need to provide more complete discussion about how their algorithm relates or does not relate to neurophysiological mechanisms.

3) The authors claim that a major factor that allows them to better investigate the origins of Fitt's Law is their physiologically realistic model of the upper limb. However the authors do not make any attempts to investigate which aspects of the model specifically are responsible, either alone, or in combination, for the observed effects. An analysis of the sort described by Lillicrap and Scott (2013, Neuron) would seem helpful here.

4) The description of the approach lacks important details. CMA-ES with a population size of 20 was used but it is not clear what parameter settings are used for sampling and how these settings influence the results. The authors refer to the solutions of the CMA-ES algorithm as local optima but this may not be mathematically correct as the number of iterations was predefined and optimality criteria were not evaluated. Also, it seems that the target was not successfully reached in all trials. For example in Figure 1B, the most rightward target is not reached. Finally, muscle excitations are prescribed every 100 ms only. This limits the set of feasible solutions. Would the results be different when increasing the number of controls (which might be more realistic physiologically)?

5) Methods related to the NHP work are lacking. In Figure 4, the authors talk about not using spike sorting and just using threshold crossings. Why? What are the implications of this approach. In general, methods associated with the monkey data are not described in the Materials and methods section. This needs substantial work to be acceptable.

[Editors' note: further revisions were suggested prior to acceptance, as described below.]

Thank you for resubmitting your article "High-fidelity musculoskeletal modeling reveals a motor planning contribution to the speed-accuracy tradeoff" for consideration by *eLife*. Your revised article has been reviewed by three peer reviewers, including Andrew Pruszynski as Reviewing Editor and Reviewer #1, and the evaluation has been overseen by Ronald Calabrese as the Senior Editor. The following individual involved in review of your submission has agreed to reveal their identity: Friedl de Groote (Reviewer #3).

The reviewers have discussed the reviews with one another and the Reviewing Editor has drafted this decision to help you prepare a revised submission.

Summary:

As you will see, the reviewers all thought that you did a very good job addressing their concerns and remain very enthusiastic about your work. There are however two important issues that remain outstanding, which the reviewers all believe need addressing to ensure the paper provides the advance it hopes to provide.

1) Previous concern #3 was not addressed and is seen by all reviewers as being very important for interpreting the findings. Specifically, the comparison is still limited to the torque-driven model and the fully complex model. The reviewers believe that it would be feasible to follow the approach of Lillicrap and Scott (2003) to assess which aspects of the model are responsible for the observed effect and that the insights gained from such assessment would be very valuable.

2) There remains a disconnect between the results of the modeling exercise and the neurophysiological data - a gap that one might expect to be bridged by a theory or model, which is still absent. In the simulations, the model behaves according to Fitt's Law because built into the model is a sort of explicit "search" within a subspace, a search that takes time. In the model this search happens pre-movement during what the authors call a movement planning epoch. The neural data are centered around a pre-movement time epoch as well, but clearly the brain is not literally executing an algorithmic "search" during this time. The reviewers all believe that, for this paper to have the desired impact, more discussion is needed to bridge the gap between the model and the nervous system.

---

## [Author Response]

Essential revisions:1) All the reviewers raised substantial concerns about how the authors handle physiological noise in their simulations. There were two related issues.a) In the present version of the model, muscle activation terms are squared and the authors suggest that this serves as an approximation of metabolic power consumption. The reviewers raised the point that, in other contexts, minimizing squared muscle activations has been interpreted as minimizing motor noise (i.e. acts like the kind of signal-dependent noise term). It is important that the authors clarify how this term influences the ability to make a clean separation between the hypotheses being tested.

This is a good point, and we thank the reviewers for raising it. Our cost function detailed in Equation 1, includes the sum of muscle activations squared. Including this term impacts the speed of a synthesized reaching movement to a target, but it does not explain the empirical observation that movements are slower to smaller targets. Simply adding the muscle activations term (without signal-dependent noise or motor planning variability) would predict the same movement duration for a large and a small target for the same reach distance (i.e., if the smaller target is placed at the edge of the larger target). Yet, Fitts’ law predicts slower movements to smaller targets. We verify these considerations by performing additional experiments to show that Fitts’ law emerges even without the muscle activations term (setting w2 to 0 in Equation 1; see subsection “Speed-Accuracy Tradeoff and Fitts’ law”). We found that the model’s mean predictions are in close agreement with Fitts’ law (R2 = 0.955). We added the figure in the manuscript as (Figure 3—figure supplement 3).

“Our cost function (Equation 1) includes a term to minimize the sum of muscle activations squared, which impacts the speed of reaching movements (but unlike signal-dependent noise, it does not predict different speeds based on target size). We performed an additional experiment (Figure 3—figure supplement 3) to show that Fitts’ law emerges even without this term in the cost function.”

b) Physiological noise is an established phenomenon that is not included in the present model. The reviewers understand why the authors chose this route (both conceptually and practically) but thought it would be important to show what happens when physiological noise is included. In this case, would the same optimization procedure with the same cost function converge similarly on a Fitts law-like solution? How would the optimized control signals in this case differ from the ones they found presently?

We appreciate having the opportunity to discuss this issue in more detail. Adding signal-dependent noise with the same cost function and the same optimization procedure (that includes motor planning variability) would converge on a Fitts’ law-like solution. The signal-to-noise ratio in the signal-dependent noise would limit how fast the reach to the target can be achieved. Hence, the signal-dependent noise would simply increase the movement duration, but the resulting curves would still be in agreement with Fitts’ law. Motor planning variability would still cause the movement duration variability to increase for smaller targets. Improvements that occur with practice could be explained either by improvements in the signal-to-noise ratio (which is not yet demonstrated experimentally), improvements in the motor plan itself (which has been demonstrated experimentally), or a combination of the two.

Adding signal-dependent noise would cause the optimized control signals to be smaller when reaching to the smaller target, but not to the larger target. This readily illustrates the difference between signal-dependent noise and the muscle activations term discussed in 1.a (which impacts the movements to both the smaller and larger targets). Adding signal-dependent noise alone without motor planning variability would not explain the behavior shown in Figure 3C or the neural data shown in Figure 4D-E.

“We exclude signal-dependent noise in our model to demonstrate that it is not necessary to explain the speed-accuracy tradeoff. Adding signal-dependent noise with the same cost function and optimization procedure (that includes motor planning variability) would also yield Fitts’ law-like solutions, but the signal-to-noise ratio would limit how fast the reach to the target can be achieved. One difficulty with modeling signal-dependent noise is that there is very little experimental data on the signal-to-noise ratio and how it changes with practice — if it changes at all. Future work exploring tasks where subjects learn a task across sessions could be quite useful in making more progress towards answering this question.”

2) All the reviewers raised the substantial concern that the present work does not establish a meaningful link between the optimal control model being proposed – and its reasons for showing Fitts' Law – and a sensible neurophysiological analog. That is, what neurally feasible computational mechanism occurs during movement preparation that produces greater variability for small targets. Given the present model algorithm does not seem feasible, what to the authors have in mind? At the very least, the authors need to provide more complete discussion about how their algorithm relates or does not relate to neurophysiological mechanisms.

We thank the reviewers for highlighting that our manuscript did not address these questions. First, prior to this study, there was little evidence to suggest any relationship between Fitts’ law and the preparatory state in motor cortex. We have now presented, to our knowledge, the first empirical evidence that such a relationship exits. That is, reaching to smaller targets exhibit less variability in the distribution of preparatory states in premotor cortex (Figure 4D); at the same time, for smaller targets a larger fraction (and the majority) of the trial-by-trial variability is explained by variability in the preparatory state (compared to larger targets; Figure 4E). Thus Fitts’ law is not entirely a result of signal-dependent noise, but also a function of neural variability during motor preparation. This is a new neurophysiological finding, which forms the basis of our proposed theory (see next paragraph).

We attempt to further investigate variability during preparation by engaging a computational model that by virtue of stochasticity during motor planning yields movement durations consistent with Fitts’ law. In particular, the model implements motor planning stochasticity with CMA-ES, which operates by applying gaussian noise on an initial motor plan.

We recognize this is not the source of neural variability during preparation in (monkey) motor cortex, but it could be as simple as noise on an initial motor plan (e.g., due to variability in the inputs from other brain regions [Vyas et al., 2020b]). We have approximated the trial-to-trial variability in the preparatory state in our model via sampling from a gaussian distribution. Although we cannot describe the exact neurophysiological mechanism during preparation that produces differential variability when reaching to smaller vs larger targets, our contribution lies in *discovering* this relationship. Furthermore, the motor plan in the model is not exactly the same as the preparatory state in premotor and primary motor cortex; they are related, but the plan encodes the full trajectory, whereas the preparatory state encodes a subset of the parameters of the upcoming movement (e.g., Messier and Kalaska, 2000; Even-Chen et al., 2019).

We also cannot assert that stochasticity in the model’s motor plan is a perfect proxy for variance in the motor cortical preparatory state, but our working hypothesis is that there is enough alignment for our theory to hold given that our simulations match well with experimental data (see subsection “Speed-Accuracy Tradeoff and Fitts’ law”); future work will need to be done to explore the extent of this alignment.

With these caveats and model assumptions in mind, the key theory being put forward here is that variability in the motor plan yields multiple candidate control solutions (Figure 5). There is less variability (Figure 4D) and fewer effective solutions when reaching to smaller targets, which yields, on average, slower reaches (see Figure 5 caption for a more thorough account of why this holds). We do not have causal evidence linking neural variability during preparation and Fitts’ law (although we have a causal link between variability during preparation and Fitts’ law in our computational model). These are important caveats and considerations regarding our theory, and we apologize for not including them in the previous version of our manuscript; this has now been addressed in the main text (as shown below).

“Our results here (Figure 4D), and those by Churchland and colleagues (Churchland et al., 2006a, 2006b, 2006c), point to preparatory activity as also being a central source of movement variability, rather than execution noise alone being the key factor. […] This provides a concrete avenue for future studies to further investigate the relative contribution of signal-dependent noise and motor planning in giving rise to the speed-accuracy tradeoff.”

3) The authors claim that a major factor that allows them to better investigate the origins of Fitt's Law is their physiologically realistic model of the upper limb. However the authors do not make any attempts to investigate which aspects of the model specifically are responsible, either alone, or in combination, for the observed effects. An analysis of the sort described by Lillicrap and Scott (2013, Neuron) would seem helpful here.

We are grateful to the reviewers for highlighting how we can improve our manuscript. We have revised the discussion to describe how a realistic musculoskeletal model allows us to better investigate the speed-accuracy tradeoff. We have shown that both a realistic musculoskeletal model and a torque-driven model produce mean movement durations that are consistent with Fitts’ law. The realistic model synthesizes movement duration means and standard deviations that are statistically correlated with experimental data (Figure 3A). The torque-driven model, however, synthesizes predicted means and standard deviations that are not statistically correlated with the experimental data. The reason is that a realistic biomechanical model allows us to more accurately estimate the optimization landscape involved in human reaching. A torque-driven model, on the other hand, can synthesize unrealistic movements (i.e., produce torques that would not have been produced by the muscles, either because it would not have been possible physiologically, or because it would have been ineffective with respect to the cost function Equation 1). More accurately estimating the optimization landscape allows us to reproduce the increase in the movement duration variability that occurs when reaching for smaller targets (Figure 3C), which we did not reproduce in our torque-driven model (Figure 3—figure supplement 1). Lastly, a torque-driven or realistic musculoskeletal model with signal-dependent noise and no planning variability produces results consistent with Fitts’ law but does not make any prediction on how movement duration variability changes with the index of difficulty. We have included a table (Author response table 1) to summarize our results.

**Author response table 1. resptable1:** Model Parameters.

	Fitts’ Law	Behavior Results Correlated with Experimental Data (movement duration means and standard deviations)
Torque or Musculoskeletal Model with Signal-Dependent Noise and without Planning Variability	Yes	No
Torque Model with Planning Variability	Yes	No
Musculoskeletal Model with Planning Variability	Yes	Yes

“We posit that a realistic biomechanical model, as described here, allows for more accurate estimation of the optimization landscape (i.e., how the local optima are situated with respect to the control solutions). This explains why the realistic biomechanical model produce results that are strongly correlated with prior experimental findings, while the simplified torque-driven model does not (since it is more likely to synthesize unrealistic movements).”

4) The description of the approach lacks important details. CMA-ES with a population size of 20 was used but it is not clear what parameter settings are used for sampling and how these settings influence the results. The authors refer to the solutions of the CMA-ES algorithm as local optima but this may not be mathematically correct as the number of iterations was predefined and optimality criteria were not evaluated. Also, it seems that the target was not successfully reached in all trials. For example in Figure 1B, the most rightward target is not reached. Finally, muscle excitations are prescribed every 100 ms only. This limits the set of feasible solutions. Would the results be different when increasing the number of controls (which might be more realistic physiologically)?

a) The reviewer is correct that the solutions of CMA-ES may not be local optima, and we revised the terminology in the paper to refer to effective/ineffective “control solutions”. We have also revised the manuscript to include the details requested by the reviewers, including all the parameters in CMA-ES. Our optimization does not require careful initialization: we set all the muscle excitations to 0.1 and all the standard deviations in the normal distribution to 0.3. We performed the following additional experiment to empirically verify that the optimization converges to similar solutions when setting the muscle excitations to 0.3 and the standard deviations to 0.5. In an average of ten trials, the average movement duration difference between the two CMA-ES parameters were 0.03 s and 0.0005 s for the smallest and largest targets (i.e., small values compared to the predicted mean movement durations in Figure 3C). We also empirically found that the movement would not successfully reach the target if the standard deviations are set too low (e.g., 0.025) or too high (e.g., 1.5). This indicates that our results are not sensitive to small changes to the reported CMA-ES parameters. It might be desirable to choose thousands of different initializations to further test the sensitivity of the results to these parameters; unfortunately, this would take months of computer time on a substantial computer cluster and was infeasible.

b) We updated Figure 1B to show that the rightward target can be reached (we had incorrectly specified the target size in the earlier figure).

c) We have performed the following additional experiment to show that doubling the number of controls (prescribing muscle excitations every 50 ms) does not affect the results. We performed ten reaches to the smallest (index of difficulty: 4.41) and largest targets (index of difficulty: 0.9) and compare the movement durations differences when prescribing the controls every 100 ms and every 50 ms. The average movement duration difference to the smallest target is 0.01 s and 0.0009 s for the largest target, which indicates that increasing the number of controls would not change the results in the paper (i.e., these are small values compared to the predicted mean movement durations in Figure 3C of 0.28 s and 0.08 s for the smallest and largest targets, respectively). We also added a target in between the smallest and largest targets (with an index of difficulty of 2.5) and again only noted a small average movement duration difference of 0.016 s when prescribing the controls every 100 ms and every 50 ms.

We have included the data and code, to be shared openly and freely, for all of our original and additional experiments described here on the following web site which we maintain: https://simtk.org/projects/ue-reaching/.

“We performed experiments to verify that prescribing the muscle excitations at every 0.05 s interval (instead of 0.1 s interval) do not have a significant impact on the computed movement durations.”

5) Methods related to the NHP work are lacking. In Figure 4, the authors talk about not using spike sorting and just using threshold crossings. Why? What are the implications of this approach. In general, methods associated with the monkey data are not described in the Materials and methods section. This needs substantial work to be acceptable.

We apologize for the lack of detailed methods for the NHP work. We relied too heavily on a previously published paper by one of the senior author’s group, but realize now that it is much better for this paper to be much more self-contained. Reproduced below are the new Methods for the NHP work. We have made clear when and where previously published techniques and experimental protocols were followed.

First, regarding spike sorting verses threshold crossings. In this work, we elected to use threshold crossings in lieu of spike sorting. That is, we do not assign individual spikes to individual neurons. Primarily, this allows us to consider all the data on the Utah electrode arrays, not just the well-isolated units. We (Trautmann et al., 2019) have recently established that there is a great deal of signal present on recording channels that do not contain well-isolated single units (i.e., so called “hash”). Thus, it is perhaps less biased to consider all the data that are available, rather than restrict the analysis to only well isolated single units. Nonetheless, we agree with the reviewers that this approach could have undesirable implications. In particular, it is not fully known what effect threshold crossing have on the raw estimation of the captured variance; perhaps threshold crossings yield a *more* biased estimate of the variance. However, it is worth noting that spike sorting itself is riddled with inaccuracies, especially when cells do not present with prototypical waveforms. Spike sorting also often results in large inter- and intra-operator variability (Wood et al., 2004). Regardless, any mis-estimation of the variance due to threshold crossing is not an issue for the primary claims of this paper for three key reasons. First, multi-units will not artificially affect the captured variance preferentially for any one condition; if anything, potentially lower SNR will make the two conditions appear more similar. Second, Figure 4 shows that the fraction of the trial-by-trial variability explained by preparatory state variability (for the large target) is ~0.5. This is essentially identical to the results found by Churchland et al., 2006a, where spike sorting was performed. Finally, and perhaps more importantly, we are not interested in the absolute values of the captured variance. We are primarily interested in assessing if there is a significant difference between the small and large target conditions; note in 4D-E, we used normalized variance instead of absolute variance.

“Recordings were made from the dorsal aspect of the premotor cortex (PMd) and the primary motor cortex (M1) of one male adult monkey (Macaca mulatta) who was 15 years old and weighed 16 kg at the time of these experiments. […] The analysis presented in Figure 4D was performed following the standard methods described in (Vyas et al., 2018; Even-Chen et al., 2019; Vyas et al., 2020b). The analysis presented in Figure 4E was performed following the standard methods described in Churchland et al., 2006a.”

[Editors' note: further revisions were suggested prior to acceptance, as described below.]

Revisions for this paper:1) Previous concern #3 was not addressed and is seen by all reviewers as being very important for interpreting the findings. Specifically, the comparison is still limited to the torque-driven model and the fully complex model. The reviewers believe that it would be feasible to follow the approach of Lillicrap and Scott (2003) to assess which aspects of the model are responsible for the observed effect and that the insights gained from such assessment would be very valuable.

We are grateful to the reviewers for highlighting how we can improve our manuscript. We performed additional experiments and analyses to address this concern and identify which components of the model are responsible for the observed effects. Following the approach of Lillicrap and Scott (2003), we introduce six different models: a torque-driven model, a musculoskeletal model missing 14 important muscles (e.g., deltoid anterior, deltoid posterior, infraspinatus, teres minor, etc.), a musculoskeletal model with unrealistic muscle force-length relationships, a musculoskeletal model with an unrealistic cost function, a realistic musculoskeletal model with an effective optimizer, and a realistic musculoskeletal with a less effective optimizer. For each of these models, we tested whether the predicted movement durations are in agreement with Fitts’ law and experimental data. Our results indicate that all models produce movement durations in agreement with Fitts’ law, but only the realistic musculoskeletal models with the effective and less effective optimizers synthesized movement duration means and standard deviations that are statistically correlated with experimental data (Figure 3A).

Our hypothesis is that a realistic biomechanical model and a realistic cost function allows us to more accurately estimate the optimization landscape involved in human reaching. The other models tend to synthesize unrealistic movements (i.e., produce torques that would not have been produced by the muscles, either because it would not have been possible physiologically, or because it would have been ineffective with respect to the cost function Equation 1). More accurately estimating the optimization landscape allows us to reproduce the increase in the movement duration variability that occurs when reaching for smaller targets (Figure 3C), which we did not reproduce in our torque-driven model (Figure 3—figure supplement 1).

Thus, our work points to the importance of accurately modeling the optimization landscape, which includes at least the following components: the biomechanical model, the cost function and the optimization parameters. We note that our results on the speed-accuracy tradeoff are robust to the optimization parameters (both the effective and less effective optimizers are statistically correlated with experimental data). An open problem for future work would be investigate other optimization algorithms and parameters and determine which ones are most consistent with experimental data. Lastly, from the literature (e.g., Harris and Wolpert, 1998), we know that a torque-driven with signal-dependent noise and no planning variability produces results consistent with Fitts’ law but does not make any prediction on how movement duration variability changes with the index of difficulty. We have included a table (Author response table 2) to summarize our results.

**Author response table 2. resptable2:** Impact of the Biomechanical Model. We summarize how our results vary based on the biomechanical model. We tested six different models: a torque-driven model, a musculoskeletal model missing 14 important muscles (e.g., deltoid anterior, deltoid posterior, infraspinatus, teres minor, etc.), a musculoskeletal model with unrealistic muscle force-length relationships, a musculoskeletal model with an unrealistic cost function, a realistic musculoskeletal model with an effective optimizer, and a realistic musculoskeletal with a less effective optimizer. The results for the model “without planning variability” are taken from the literature.

	Fitts’ Law	Behavior Results Correlated with Experimental Data (movement duration means and standard deviations)
Torque or musculoskeletal model with signal-dependent noise and without planning variability	Yes	No
Torque model	Yes	No
Musculoskeletal model with missing muscles	Yes	No
Musculoskeletal model with unrealistic force-length relationships	Yes	No
Musculoskeletal model with an unrealistic cost function (i.e., no muscle activations term)	Yes	No
Musculoskeletal model with an effective optimizer	Yes	Yes
Musculoskeletal model with a less effective optimizer	Yes	Yes

We have made the following changes to address these points in the manuscript. We have added Table 1 and Figure 3—figure supplement 4 and Figure 3—figure supplement 5.

In subsection “Speed-Accuracy Tradeoff and Fitts’ law”:

“We also tested the sensitivity of our results to features of the biomechanical model: we performed the experiments with a model with muscles having unrealistic active force-length relationships (i.e., where the force is constant with respect to the muscle length; Figure 3—figure supplement 4) and a model missing important muscles (e.g., deltoid anterior, deltoid posterior, infraspinatus, teres minor; Figure 3—figure supplement 5). We found that these models produced movement durations that are consistent with Fitts’ law, but the predictions were not correlated with the experimental data in Goldberg et al., 2014 (Table 1).”

In the Discussion:

“We posit that a realistic biomechanical model, as described here, allows for more accurate estimation of the optimization landscape (i.e., how the local optima are situated with respect to the control solutions). This explains why the realistic biomechanical model produces results that are strongly correlated with prior experimental findings (Goldberg et al., 2014), while all of our simplified models (which includes the torque-driven, fewer muscles, unrealistic force-length relationships, and unrealistic cost function models) are not statistically correlated with prior experimental findings since they are more likely to synthesize unrealistic movements (Table 1).”

2) There remains a disconnect between the results of the modeling exercise and the neurophysiological data – a gap that one might expect to be bridged by a theory or model, which is still absent. In the simulations, the model behaves according to Fitt's Law because built into the model is a sort of explicit "search" within a subspace, a search that takes time. In the model this search happens pre-movement during what the authors call a movement planning epoch. The neural data are centered around a pre-movement time epoch as well, but clearly the brain is not literally executing an algorithmic "search" during this time. The reviewers all believe that, for this paper to have the desired impact, more discussion is needed to bridge the gap between the model and the nervous system.

We are grateful to the reviewers for highlighting how we can improve the presentation and discussion of our theory. Our modeling exercise and the subsequent non-human primate experiments are suggestive of a new theory, one that hopefully will inspire much more future work on the speed-accuracy tradeoff. Our new proposal can be better understood if placed side-by-side with the prevailing theory (Harris and Wolpert, 1998).

Prevailing theory:

Harris and Wolpert suggest that on a trial where you reach to a small target, you need to modify your control signal to be smaller if you wish to minimize your variability in the endpoint position. This naturally slows down your reaches, thus yielding (some of) the empirical aspects of Fitts’ law. In this theory, if the neural control signal is corrupted by noise that scales with the size of the control signal, then in order to minimize endpoint variability, motor cortex issues commands with smaller overall mean to accomplish the task.

Our proposal:

At its core essence, our theory relies on motor planning variability and questions whether signal-dependent noise is the *only* factor that ultimately gives rise to Fitts’ law.

1) We start with previous findings by Churchland and colleagues that demonstrate that trial by-trial movement duration variability, even during highly practiced tasks, can be accounted for by motor planning (but more specifically preparatory) variability.

2) Given these findings, we introduce a computational model with planning variability and demonstrate its ability to accurately recapitulate many previous (and sometimes puzzling) results and phenomenon, including Fitts’ law. In our opinion, this alone is a non-trivial contribution.

a) We focus on the modeling assumption that the movement variability resulting from motor planning variability can be modeled with the solutions found from stochastic optimization (though note, we do not think that the brain literally performs a stochastic optimization during motor planning). Our model’s account of Fitts’ law does not rely on the optimization time or the explicit search before finding the motor plans. The optimization search will explore many possible motor plans before converging to a set of possible solutions. Our theory only relates to the final (converged) solutions. Our model needs to perform a search to converge to this set of possible solutions, but the central nervous system does not.

b) Our model shows that for more stringent task constraints, effective solutions are rendered inaccessible (Figure 5). When comparing the reaches to a large and a small target, we have empirically verified that the reaches to the large target that also do not reach the small target are on average faster than the reaches to the small target. This yields on average slower reaches to smaller targets. We hypothesize that this also occurs in the brain when reaching to targets of different sizes.

3) We test this hypothesis by analyzing motor cortical activity of monkey reaching movements. First, we replicate the findings by Churchland and colleagues 2006, and both extend them for harder tasks (i.e., reaching to smaller targets) and show that variability in the preparatory state in motor cortex is associated with some of the empirical aspects of Fitts’ law.

4) Having established that there is a relationship between movement duration variability and planning variability, let’s consider what happens when making reaches to a small target. Empirically, more (and the majority) of the trial-by-trial movement duration variability is accounted for by variability in the preparatory state (compared to a larger target). At the same time, there is less variability in the preparatory state in motor cortex and more movement duration variability. For reaches to a larger target, there is more variability in the preparatory state in motor cortex and less movement duration variability.

5) This new empirical evidence indicates that motor planning variability has a significant contribution to the speed-accuracy tradeoff and is consistent with our theory, which predicts that reaches to larger targets are on average faster because the brain has access to a larger set of effective motor plans compared to smaller targets.

When viewed in this manner, it is the combination of previous neurophysiology, new behavior and neurophysiology, and computational modeling that gives rise to this theory. Our proposal does not require signal-dependent noise. Our proposal explains many of the confusing looking responses seen in previous studies that signal-dependent noise cannot explain. Perhaps the reviewers would agree that this meets the bar for publication at this time. Indeed, significant future work will be needed to pin down an *exact* neurophysiological mechanism. To put it succinctly, we have proposed a new hypothesis, one that is very different from the prevailing theory, and our succinct hope is that future studies will either deepen or disprove our proposal.

We also added the following testable prediction of our theory. With practice, there’s a shift in in performance that allows movements to be performed faster and more accurately. Our theory predicts that the neural variability during motor planning should progressively decrease as the shift in performance occurs (and as the central nervous system converges to better motor plans).

We have revised the Discussion to clarify the presentation of our theory:

“Turning our attention to neural mechanisms underlying motor cortical function, a current theory suggests that movement period neural activity evolves from an initial condition (termed the neural population preparatory state) set in premotor and primary motor cortex (Shenoy et al., 2013, Vyas et al., 2020a). […] This indicates a potential role for planning variability in giving rise to the speed-accuracy tradeoff, which is supported by further neurophysiological evidence introduced here (Figure 4D and Figure 4E).”

We included our testable prediction here:

“During motor learning, there is a shift in performance that allows movements to be performed faster and more accurately. […] A testable prediction of our theory is that the neural variability during motor planning should progressively decrease as the shift in performance occurs during practice (and as the central nervous system converges to better motor plans).”